# Immediate unfavorable birth outcomes and determinants of operative vaginal delivery among mothers delivered in East Gojjam Zone Public Hospitals, North West Ethiopia: A cross-sectional study

Habtamu Sewunet[1], Nurilign Abebe[1], Liknaw Bewket Zeleke[1,2], Bewket Yeserah Aynalem[1]*, Addisu Alehegn Alemu[1]

1 Debre Markos University, Debre Markos, Ethiopia, 2 School of Women's and Children's Health, University of New South Wales Sydney, Sydney, Australia

* by123bewket@gmail.com

**Data Availability Statement:** All relevant data are within the paper and its Supporting Information files.

## Abstract

### Introduction

Operative vaginal delivery is the use of forceps or vacuum devices to assist the eligible laboring mother to avoid poor birth outcomes. It is associated with increased maternal, neonatal morbidity and perinatal complications if it is not used appropriately. Instrumental delivery use needs health care providers' skills, knowledge, and decision-making ability for good maternal outcomes.

### Objective

This study aimed to assess immediate unfavorable birth outcomes and associated factors of operative vaginal delivery among women delivered in East Gojjam Zone Public Hospitals, North West Ethiopia.

### Method

The study design was institution based cross-sectional and consecutive sampling procedure was used to select 313 mothers in the study, from March 1, 2019, to April 30, 2019. We used Epi data version 3.1 for data entry and SPSS version 25 software for cleaning and analysis. A Bivariable logistic regression analysis was used to identify the association between each outcome variable and each factor. Again, a multivariable logistic regression analysis was employed to identify factors associated with each outcome variable, and variables with a p-value less than 0.05 were taken as significant variables.

### Results

The overall unfavorable maternal outcomes of operative vaginal delivery were found to be 32.9% [95% CI: 27.8, 38.3]. No formal education (AOR = 8.36; 95% CI: 1.01, 69.2), rural residence (AOR: 11.77; 95% CI: 2.02, 68.41), male sex of the neonate (AOR: 2.87; 95% CI:

**Abbreviations:** NICU, neonatal intensive care unit;
OVD, Operative vaginal delivery; PPH, Postpartum
hemorrhage; SSOL, shortening of the second stage
of labor.

1.08, 7.61) and zero station during instrumental application (AOR: 6.93; 95% CI: 1.75, 27.5) were factors associated with unfavorable maternal outcomes. The study also showed that the magnitude of unfavorable neonatal outcomes was 34.8% (95% CI: 29.7, 40.3). Vaginal first-degree tear (AOR = 0.03, 95% CI: 0.001, 0.951) and blood transfusion (AOR = 7.38, 95% CI: 1.18–46.15) was statistically significant factors associated with unfavorable neonatal outcomes.

## Conclusion

The overall unfavorable maternal and neonatal outcomes of operative vaginal delivery were high compared with some other studies done in Ethiopia.

## Introduction

Operative vaginal delivery (OVD) is the use of forceps or vacuum devices to assist the eligible laboring mother to avoid poor birth outcomes [1, 2]. The skill of health care providers, decision-making ability, and knowledge about the indication of instrumental delivery were the contributing factors for poor immediate birth outcomes [3–6].

OVD is a key element of essential obstetric care in resource-poor countries which reduces maternal and neonatal mortality/morbidity if it is used appropriately with small manpower and minimized supply resources [7]. Forceps delivery was is associated with increased maternal, neonatal morbidity, and perinatal complications in previous studies [8]. In a study done in Jimma university medical center, southwest Ethiopia, 3.3% were complicated with Postpartum hemorrhage (PPH) due to uterine atony and episiotomy extension cases like fourth-degree genital tear [9, 10]. The poor neonatal outcome is also associated with the skill and knowledge of the health care providers [11]. In Ethiopia, in-service training is being given on operative vaginal delivery as part of basic emergency management obstetric care to improve the experience of health care providers [11].

There is substantial evidence that instrumental deliveries increase maternal morbidity, pain at delivery, pain in the immediate postpartum period, perineal lacerations, hematomas, blood loss and anemia, urinary retention, and long-term problems with urinary and fecal incontinence [12]. Urinary and anal dysfunction (including incontinence, fistula formation, and pelvic organ prolapse) are additional risks of instrumental delivery that typically present months to years after delivery [10, 13]. The inaccessibility and unavailability of health facilities, instruments that are important for assisted delivery service, and nearby health facilities that aren't equipped with qualified personnel and instruments increased the risk of fetal and maternal mortality and morbidity [9]. Despite the significant burden of OVD, no evidence is done on the outcome and factors associated with OVD in the study area. Thus this study aimed to assess immediate unfavorable birth outcomes and associated factors of Operative Vaginal Delivery among women delivered in East Gojjam Zone Public Hospitals, North West Ethiopia.

## Methods

### Study setting and period

East Gojjam zone is found in Amhara national regional state, northwest Ethiopia and the administrative center of the zone is Debre Markos town. As the 2007 census conducted by the

central statistical agency indicated, a total of 2,153,937 population lives in the zone of which 1,087,221 are females. The zone has nine hospitals that provide different health services. The study was conducted from March 1, 2019, to April 30, 2019.

## Study design

An institution-based cross-sectional study was conducted.

## Study participants

The source population was mothers who gave birth assisted with instrumental delivery in East Gojjam Zone Public Hospitals and mothers who gave birth assisted with instrumental delivery in East Gojjam Zone Public Hospitals during the data collection time were the study population. Mothers who gave birth with gross congenital anomalies and were referred from nearby health institutions after attempted instrumental delivery were excluded.

## Sample size and sampling procedure

The sample size was calculated based on a single population proportion formula assumption. The proportion of unfavorable birth outcomes was 0.24 from a study conducted in Arbaminch General Hospital, Southern Ethiopia [4] with a 5% margin of error.

$$\text{initial sample size} = \left(Z\frac{a}{2}\right)^2 * \frac{p(1-p)}{w2} = 1.96^2 * \frac{0.24(1-0.24)}{(0.05)^2} = 281$$

By considering 10% non-response rates, the final sample size was 313.

## Sampling procedure

The study participants were selected from all hospitals and samples were proportionally allocated for each hospital. Then the required samples were taken by using consecutive sampling techniques. The data were collected immediately after giving birth when the women were comfortable before leaving the hospital and the immediate feto-maternal birth outcomes were assessed.

## Study variables
## Dependent variable:

- Immediate unfavorable birth outcome

## Independent variables/explanatory variables

**Socio-demographic variables:** age, residence, marital status, religion, educational level, family monthly income, ethnicity

**Obstetrics and gynecologic variables:** gravidity, parity, number of antenatal care visits, gestational age, duration of labor, cervical dilatation

**Health professional related factors:** experience of professionals, qualifications of professionals

**Fetal/neonate related factors:** sex of neonate, station, position, weight

## Operational definitions

**Unfavorable maternal Outcome:** The unfavorable outcome is when the mother developed one or more complications among the following complications: postpartum hemorrhage,

genital tear, fistula, need of blood transfusion, need of major surgery, death related to instrumental delivery (10).

**Unfavorable neonatal Outcome** is considered when the neonate experience one or more from the following complication: asphyxia, birth injuries, retinal hemorrhage, subgalial hemorrhage, cephalo-hematoma, anemia, need of resuscitation, admission to neonatal intensive care unit (NICU), and neonatal death (10).

**Immediate birth outcomes** are the immediate maternal or neonatal conditions that could be complicated or non-complicated occurring within the first six hours of delivery [9].

### Data collection tool and data quality control

An interviewer-administered structured, face-to-face observation of outcomes and patient chart review and pre-tested questionnaire was used by 20 trained BSc midwives and 2 masters of public health professionals. Two-day data collection training was given for both data collectors and supervisors. A pretest was conducted on 5% of the sample size in Finote-Selam hospital and the necessary correction was taken accordingly.

### Data processing and analysis

The collected data were entered into Epi data version 3.1 and then exported to SPSS version 25 for data cleaning and analysis. Descriptive statistics were computed to describe the study population about relevant variables. Binary logistic regression was used to identify factors associated with the outcome variable. Independent variables that showed P-value $\leq 0.25$ in the bivariate logistic regression analysis were included in the multivariable logistic regression analysis. Finally, variables with P-value $< 0.05$ at a 95% confidence interval were declared as significantly associated with the outcome variable. The strength & direction of the association was interpreted based on the adjusted odds ratio.

**Ethical clearance.**   The Ethical Review Committee of Debre Markos University Health Sciences College approved the ethical acceptability of the study. Then the ethics approval letter was obtained from the university and submitted to the hospitals. Written informed consent was obtained from each woman after explaining the purpose and ethical process of the study. Finally, the women were interviewed in private rooms independently and the data were kept anonymous.

## Results

### Socio-demographic characteristics

All the intended 313 women delivered through operative vaginal birth participated in the study making the response rate 100%. The age range of the study participants was between 18 and 48 years with the mean age and standard deviation of 27 ± 5.833 SD years. Nearly one-third (32.9%) participants were found in the age group of 25 to 29 years. Most of the study participants were Orthodox Christian followers (86.6%) and participants were married (91.1%). Concerning the educational level and occupation, about one-third had no formal education (35.1%) and farmers (32.6%). More than half of the participants' monthly income was greater than 1742 Ethiopian Birr (ETB) (58.1%) and urban residents (57.2%) from a residence perspective (Table 1).

### Obstetric and related characteristics

The study showed that nearly half (47.9%) of the participants had given birth to their first child in terms of their parity and 91.7% of the participants had antenatal care follow-up. Nearly half of 154(49.2%) participants have ultrasound investigations during pregnancy. The majority

**Table 1. Socio-demographic characteristics of the study participants on OVD at East Gojjam Zone Hospitals, Northwest Ethiopia, 2019(N = 313).**

| Variables | Number | Percentage |
|---|---|---|
| **Age in Years** | | |
| 15–19 | 15 | 4.8 |
| 20–24 | 89 | 28.4 |
| 25–29 | 103 | 32.9 |
| 30–34 | 58 | 18.5 |
| Above 35 | 48 | 15.3 |
| **Religion** | | |
| Orthodox | 271 | 86.6 |
| Muslim | 42 | 13.4 |
| **Marital status** | | |
| Single | 17 | 5.4 |
| Married | 285 | 91.1 |
| Divorced | 11 | 3.5 |
| **Educational status** | | |
| No formal schooling | 110 | 35.1 |
| Primary level (1–8) | 75 | 24.0 |
| Secondary level (9–12) | 56 | 17.9 |
| College and above | 72 | 23.0 |
| **Occupation** | | |
| Farmer | 102 | 32.6 |
| Merchant | 60 | 19.2 |
| Government employee | 61 | 19.5 |
| Private employee | 36 | 11.5 |
| Housewife | 54 | 17.3 |
| **Residence** | | |
| Urban | 179 | 57.2 |
| Rural | 134 | 42.8 |
| **Income in Ethiopian Birr** | | |
| Less and equal 500 | 34 | 10.9 |
| 501–1742 | 97 | 31 |
| Greater than 1742 | 182 | 58.1 |

(92.3%) of the respondents have given birth at 37 to 42 weeks in terms of gestational age by the last menstrual period. More than half (59.1%) of study participant mothers' weight was found below 60 kg and nearly one-third (36.1%) of respondents' mid-upper arm circumference was greater than 23. Greater than 77% of the respondents were assisted by midwives. Seventy-seven percent of health care providers who performed vaginal operative delivery experience were less than five years. The commonest indication for OVD was poor maternal effort (51.1%). More than three-fourths of 241(77%) applied instruments on Occipto anterior position, among the types of OVDs Vacuum is more commonly used 232(74.1%). Almost all vacuum applied were plastic 231(99.6%) and more than half 171(54.6%) of the participants were delivered without episiotomy (Table 2).

## Unfavorable feto-maternal outcomes

The study revealed that nearly one-third (32.9%) of women have developed unfavorable maternal outcomes resulting from operative vaginal delivery. About one out of ten (10.5%)

women experienced postpartum hemorrhage, and 31.3% developed perineal tear due to the procedure.

Concerning immediate neonatal outcomes, one hundred nine (34.8%) neonates developed unfavorable favorable neonatal outcomes. The most frequently experienced unfavorable neonatal outcome was asphyxia (39.9%) followed by neonatal resuscitation (32.3%).

### Factors associated with unfavorable maternal outcomes of OVD

Factors associated with unfavorable feto-maternal outcomes were identified by binary logistic regression analysis. All independent variables were explored for their eligibility for logistic regression and eligible variables were then used for the analysis. Variables with a *p*-value <0.25 in the bivariate logistic regression were selected for multivariable logistic regression. A *p*-value <0.05 was used to declare a significant association in the multivariable binary logistic regression.

In bivariable binary logistic regression; educational status, residence, income, gravidity, parity, number of antenatal care, the weight of the mother, experience of professional, prolonged 2nd stage labor, position, type of instrument, and sex of neonate were found to be eligible for multivariable binary logistic regression. Finally, educational level, residence, sex of neonate, and station were found to be significantly associated with maternal outcomes of OVD (P-value<0.05) were identified associated factors with the unfavorable maternal outcomes.

Educational level [AOR: 8.36 CI (1.01, 69.22)], resident [AOR: 11.77, CI (2.02, 68.41), sex of neonate [AOR: 2.87, CI (1.08, 7.61)] and station [AOR: 6.93(1.75–27.48) were the significant factors for unfavorable maternal outcome (Table 3).

### Factors associated with unfavorable fetal outcomes of OVD

Similarly, factors associated with unfavorable neonatal outcomes of OVD were identified through bivariable and multivariable logistic regression analysis. In bivariable analysis, antenatal care follow-up, non-reassuring fetal heart rate pattern, prolonged second stage, vaginal tear, blood transfusion, and weight of neonate showed association with unfavorable neonatal outcomes of OVD. In multivariable logistic regression, vaginal tear [AOR: 0.031, CI (0.001, 0.951)] and blood transfusion [AOR: 7.38, CI(1.18,46.15)] for the mother were found to be significantly associated with fetal outcomes of OVD (P-value<0.05) (Table 4).

### Discussion

The finding of this study showed that the magnitude of unfavorable maternal outcomes is found to be high (32.9%) when compared with a study done conducted in Jimma [9]. The possible explanation could be due to skill differences with the operators, socio-demographic status of women, less antenatal care follow-up in the current study.

Women who could not read/write were eight times more likely to develop unfavorable maternal outcomes during operative vaginal delivery as compared with women whose educational statuses were college and above. This might be due to those who can't read and write may not have the ability to decide on their own; which means whenever the mother is educated, her level of understanding about the function of health care services might be improved which makes them start antenatal care follow up as soon as possible and may have continuous follow up that may help to detect and reduces the complications early.

Those who were from rural were about twelve times more likely to develop unfavorable maternal outcomes than their counterparts. This study was supported by a study done in Suhul, Tigray, Ethiopia [14]. This might be due to mothers from the rural area may not have

**Table 2. Obstetric and related characteristics of operative vaginal delivered mothers in East Gojjam Zone Hospitals, Northwest Ethiopia, 2019.**

| Variables | Number | Percentage |
| --- | --- | --- |
| Gravida | | |
| Primi gravida | 138 | 44.1 |
| Multigravida | 175 | 55.9 |
| **Parity** | | |
| 1 | 150 | 47.9 |
| 2–4 | 146 | 46.6 |
| ≥5 | 17 | 5.4 |
| **Gestational age (LMP)** | | |
| <37 weeks | 19 | 6.1 |
| 37–42 weeks | 289 | 92.3 |
| ≥42 weeks | 5 | 1.6 |
| **ANC follow up** | | |
| Yes | 287 | 91.7 |
| No | 26 | 8.3 |
| **First antenatal care visit** | | |
| <12 | 128 | 40.9 |
| 12–24 | 128 | 40.9 |
| 24–37 | 31 | 9.9 |
| **No of antenatal care visit** | | |
| <3 | 80 | 25.6 |
| 3–5 | 193 | 61.7 |
| **Current pregnancy Complications** | | |
| Yes | 71 | 22.7 |
| no | 242 | 77.3 |
| **Weight (Kg)** | | |
| < 60 | 185 | 59.1 |
| ≥ 60 | 128 | 40.9 |
| **mid-upper arm circumference** | | |
| <21 | 104 | 33.2 |
| 21–23 | 96 | 30.7 |
| >23 | 113 | 36.1 |
| **OVD performer's profession** | | |
| Midwife | 242 | 77.3 |
| IESO | 58 | 18.5 |
| Obstetrician | 13 | 4.2 |
| Duration of $2^{nd}$ stage primipara (n = 134) | | |
| <120minutes | 92 | 29.4 |
| ≥120minutes | 42 | 13.4 |
| Duration of $2^{nd}$ stage multipara (n = 179) | | |
| <60 | 88 | 28.1 |
| > = 60 | 91 | 29.1 |
| Cervical dilatation during the procedure | | |
| <10cm | 7 | 2.2 |
| 10cm | 306 | 97.8 |
| Station when instrument application | | |
| Outlet | 23 | 7.3 |

(*Continued*)

**Table 2.** (Continued)

| Variables | Number | Percentage |
|---|---|---|
| Low (station+2, +3) | 116 | 37.1 |
| Mid (station0, +1) | 174 | 55.6 |
| Position when instrument application | | |
| Occiptoanterior | 241 | 77.0 |
| Occiptolateral | 39 | 12.5 |
| Occiptoposterior | 33 | 10.5 |
| After coming head | | |
| Yes | 3 | 0.3 |
| No | 320 | 99.7 |
| Types of instruments used | | |
| Forceps | 72 | 23.0 |
| Vacuum | 232 | 74.1 |
| Both | 9 | 2.9 |
| Number of Pulls applied | | |
| <3 | 304 | 97.1 |
| ≥3 | 9 | 2.9 |
| Duration of Procedure | | |
| 0–29 minutes | 243 | 77.6 |
| 30–59 minutes | 67 | 21.4 |
| ≥60 minutes | 3 | 1.0 |
| Episiotomy done | | |
| Yes | 142 | 45.4 |
| No | 171 | 54.6 |

the information and the level of education may not be good as mothers from the urban area so that the possibility of getting antenatal care to follow will be less. Again the other reason may also be there is a great difference in the accessibility of health facility, and transportation, which makes it difficult to get the appropriate services on time, and their socio-economic difference make them weak during the OVD as compared with their counterparts.

Male sex of neonate was an associated factor for unfavorable maternal outcomes of operative vaginal delivery. Those women who deliver male neonates were 2.9 times more likely to develop unfavorable maternal outcomes of operative vaginal delivery compared with women who delivered female neonates. The reason might be male fetus has the probability of being post-term pregnant [15] and as a result, weight might be increased than female fetus which increases the bad maternal outcome of operative vaginal delivery.

Those respondents who had station zero during instrument application were seven times more likely to develop unfavorable maternal outcomes of operative vaginal delivery compared with instruments applied at station +3. This finding is supported by other studies done in Jimma, Ethiopia [16] and Suhul Tigray Ethiopia [14].

The possible explanation could be; if the station is high the possibility to identify a fetal position and to apply instruments could be difficult as compared to those who had station 3+; again when the station is high, it needs experienced expertise operator that may not available in all health institutions so that it may lead poor maternal outcome.

The finding of this study showed that the magnitude of unfavorable neonatal outcomes is found to be 34.8%. The finding of this study is higher than the study conducted in Jimma [9], Arbaminch [4]. The possible difference might be the difference in skill of the health care

**Table 3. Factors associated with unfavorable maternal outcomes of operative vaginal delivery in East Gojjam Zone Hospitals, Northwest Ethiopia.**

| Variables | Maternal Outcome | | COR (95% CI) | AOR (95% CI) |
|---|---|---|---|---|
| | Unfavorable | Favorable | | |
| **Education** | | | | |
| No formal schooling | 23 | 87 | 0.37(0.19, 0.71) | 8.36(1.01, 69.22) * |
| Primary (1–8) | 26 | 49 | 0.74(0.38, 1.45) | 1.02(0.19, 5.29) |
| Secondary (9–12) | 24 | 32 | 1.05(0.52, 2.129) | 0.89(0.15, 5.23) |
| College and above | 30 | 42 | 1 | 1 |
| **Residence** | | | | |
| Urban | 70 | 109 | 1 | 1 |
| Rural | 33 | 101 | 0.51(0.31, 0.83) | 11.77(2.02, 68.41)* |
| **Income** | | | | |
| ≤500 | 8 | 26 | 1 | 1 |
| 501–1742 | 26 | 71 | 1.08(0.41–2.84) | 3.05(0.42, 22.11) |
| ≥1743 | 69 | 113 | 1.91(0.82–4.42) | 0.21(0.02, 2.03) |
| **Gravida** | | | | |
| Primigravida | 55 | 83 | 1.75(1.08–2.82) | 1.41(0.39, 5.11) |
| Multigravida | 48 | 127 | 1 | 1 |
| **Parity** | | | | |
| 1 | 59 | 91 | 4.92(1.09–22.29) | 1.94(0.26, 14.29) |
| 2–4 | 42 | 104 | 3.03(0.66,13.83) | 1.89(0.39, 9.16) |
| > = 5 | 2 | 15 | 1 | 1 |
| **Number of ANC** | | | | |
| <3 | 19 | 61 | 0.13(0.35,0.44) | 0.02(0.00,1505.82) |
| 3–5 | 65 | 128 | .20(0.06,0.67) | 0.01(0.00,914.31) |
| >5 | 10 | 4 | 1 | 1 |
| **Weight of the mother** | | | | |
| lessthan60kg | 56 | 129 | 0.75(0.46,1,21) | 0.35(0.09,1.31) |
| 60-80kg | 47 | 81 | 1 | 1 |
| **Experience** | | | | |
| <5years | 72 | 169 | 0.56(0.33,0.97) | 0.50(0.12, 2.17) |
| ≥5years | 31 | 41 | 1 | 1 |
| **Prolonged 2nd stage labor** | | | | |
| Yes | 57 | 78 | 2.12(1.32,3.43) | 0.68(0.24, 1.88) |
| No | 46 | 132 | 1 | 1 |
| **Position** | | | | |
| Occipto-anterior | 75 | 166 | 1 | 1 |
| Occipto-lateral | 20 | 19 | 2.33(1.18–4.62) | 1.10(0.19, 6.16) |
| Occipto-posterior | 8 | 25 | 0.71(0.31–1.64) | 3.48(0.37, 32,70) |
| **Sex of neonate** | | | | |
| Male | 63 | 112 | 1.38(0.85,2.23) | 2.87(1.08,7.61) * |
| Female | 40 | 98 | 1 | 1 |
| **Neonatal Resuscitation** | | | | |
| Yes | 43 | 58 | 1.88(1.15,3.08) | 1.67(0.63,4.42) |
| No | 60 | 152 | 1 | 1 |
| **Referred to NICU** | | | | |
| Yes | 27 | 27 | 2.41(1.33,4.37) | 1.02(0.17,6.12) |
| No | 76 | 183 | 1 | 1 |
| **Procedural trauma** | | | | |

(*Continued*)

**Table 3.** (Continued)

| Variables | Maternal Outcome | | COR (95% CI) | AOR (95% CI) |
|---|---|---|---|---|
| | Unfavorable | Favorable | | |
| Yes | 38 | 22 | 4.99(2.75,9.07) | 2.04(0.41,10.15) |
| No | 65 | 188 | 1 | |
| **Station during instrument application** | | | | |
| 0 | 28 | 36 | 2.95(0.98–8.89) | 6.93(1.75–27.48) * |
| 1 | 36 | 87 | 1.57(0.55–4.53) | 2.51(0.69,9.001) |
| 2 | 34 | 68 | 1.90(0.65–5.53) | 2.86(0.79,10.20) |
| 3 | 5 | 19 | 1 | 1 |

*$p < 0.05$

providers and socio-demographic difference of the mothers which may be the reason for less antenatal care.

Developing unfavorable neonatal outcomes among mothers with first-degree tear is reduced by 96.9% as compared to mothers with a third-degree tear. The possible reason for this might be mothers who had tight perineum to pass the fetus through pelvic canal first-degree tear occur and which reduce the unfavorable neonatal outcome of operative vaginal delivery.

Those women who were blood transfused due to PPH were seven times more likely to develop unfavorable neonatal outcomes than non-transfused mothers. A possible reason might be excessive instrumental delivery due to high station and tight perineum which leads to further trauma for neonates.

**Table 4.** Factors associated with fetal outcomes of OVD in East Gojjam Zone Hospitals, Northwest Ethiopia.

| Variables | Fetal Outcome | | COR | AOR |
|---|---|---|---|---|
| | Unfavorable | Favorable | | |
| **ANC follow up** | | | | |
| No | 16 | 10 | 0.30(0.13,0.69) | 0.19(0.02,1.84) |
| Yes | 93 | 194 | 1 | 1 |
| **Nonreassuring fetal heart rate pattern** | | | | |
| Yes | 57 | 73 | 1.96(1.22,3.15) | 5.02(0.58,43.62) |
| No | 51 | 128 | 1 | |
| **Prolonged second stage labor** | | | | |
| Yes | 60 | 74 | 2.13(1.33,3.47) | 0.55(0.05,5.74) |
| No | 49 | 129 | 1 | 1 |
| **Vaginal tear** | | | | |
| First degree | 6 | 13 | 0.15(0.02,0.99) | 0.03(0.001, 0.951* |
| Second degree | 17 | 14 | 0.41(0.07,2.33) | 0.04(0.002, 1.13) |
| Third-degree | 6 | 2 | 1 | 1 |
| **Blood transfused** | | | | |
| No | 96 | 201 | 1 | 1 |
| Yes | 13 | 3 | 9.07(2.53,32.59) | 7.38(1.18, 46.15) * |
| **Weight of neonate** | | | | |
| <2.5 | 11 | 13 | 0.17(0.02,1.68) | 0.170(0.014, 2.121) |
| 2.5–4 | 93 | 190 | 0.09(0,01,0.85) | 0.14(0.02, 1.37) |
| >4 | 5 | 1 | 1 | 1 |

*$p < 0.05$

## Strength and limitations

### Strength

The study is conducted in a wide area, at the Zonal level, and using primary data makes it more representative of the sample that possibly predicts outcomes of operative vaginal delivery.

### Limitation

The study design used had a short follow-up that fails to predict long-term complications of operative vaginal delivery for the mother as well as the newborn.

### Conclusion

This study revealed that the overall unfavorable maternal and neonatal outcomes of operative vaginal delivery are found to be relatively higher in East Gojjam Zone public health hospitals than in studies done at other places in Ethiopia. Can't read and write educational status, rural residence, male sex of neonate, and station zero had a statistically significant association with unfavorable maternal outcomes of operative vaginal delivery. Vaginal first-degree tear and blood transfusion due to PPH had a statistically significant association with unfavorable neonatal outcomes of operative vaginal delivery.

## Supporting information

**S1 File.**
(DOCX)

## Acknowledgments

We would like to express our deepest heartfelt thanks to Debre Markos University for their permission to do this research and we gratefully acknowledge all study individuals for their participation in the study.

## Author Contributions

**Conceptualization:** Habtamu Sewunet, Nurilign Abebe, Liknaw Bewket Zeleke, Bewket Yeserah Aynalem, Addisu Alehegn Alemu.

**Data curation:** Habtamu Sewunet, Nurilign Abebe, Liknaw Bewket Zeleke, Bewket Yeserah Aynalem, Addisu Alehegn Alemu.

**Formal analysis:** Habtamu Sewunet, Nurilign Abebe, Liknaw Bewket Zeleke, Bewket Yeserah Aynalem, Addisu Alehegn Alemu.

**Funding acquisition:** Habtamu Sewunet, Nurilign Abebe, Liknaw Bewket Zeleke, Bewket Yeserah Aynalem, Addisu Alehegn Alemu.

**Investigation:** Habtamu Sewunet, Nurilign Abebe, Liknaw Bewket Zeleke, Bewket Yeserah Aynalem, Addisu Alehegn Alemu.

**Methodology:** Habtamu Sewunet, Nurilign Abebe, Liknaw Bewket Zeleke, Bewket Yeserah Aynalem, Addisu Alehegn Alemu.

**Project administration:** Habtamu Sewunet, Nurilign Abebe, Liknaw Bewket Zeleke, Bewket Yeserah Aynalem, Addisu Alehegn Alemu.

**Resources:** Habtamu Sewunet, Nurilign Abebe, Liknaw Bewket Zeleke, Bewket Yeserah Ayna-lem, Addisu Alehegn Alemu.

**Software:** Habtamu Sewunet, Nurilign Abebe, Liknaw Bewket Zeleke, Bewket Yeserah Ayna-lem, Addisu Alehegn Alemu.

**Supervision:** Habtamu Sewunet, Nurilign Abebe, Liknaw Bewket Zeleke, Bewket Yeserah Aynalem, Addisu Alehegn Alemu.

**Validation:** Habtamu Sewunet, Nurilign Abebe, Liknaw Bewket Zeleke, Bewket Yeserah Aynalem, Addisu Alehegn Alemu.

**Visualization:** Habtamu Sewunet, Nurilign Abebe, Liknaw Bewket Zeleke, Bewket Yeserah Aynalem, Addisu Alehegn Alemu.

**Writing – original draft:** Habtamu Sewunet, Nurilign Abebe, Liknaw Bewket Zeleke, Bewket Yeserah Aynalem, Addisu Alehegn Alemu.

**Writing – review & editing:** Habtamu Sewunet, Nurilign Abebe, Liknaw Bewket Zeleke, Bewket Yeserah Aynalem, Addisu Alehegn Alemu.

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
