## [Decision Letter · Decision Letter 0]

21 Jan 2022

PONE-D-21-32649R1

Immediate Unfavorable Birth outcomes and determinants of Operative Vaginal Delivery among Mothers Delivered in East Gojjam Zone Public Hospitals, North West Ethiopia: a cross-sectional study

PLOS ONE

Dear Dr. Aynalem,

Thank you for submitting your manuscript to PLOS ONE. After careful consideration, we feel that it has merit but does not fully meet PLOS ONE’s publication criteria as it currently stands. Therefore, we invite you to submit a revised version of the manuscript that addresses the points raised during the review process.

We look forward to receiving your revised manuscript.

Kind regards,

Sarah Silva

Academic Editor

PLOS ONE

Reviewers' comments:

Reviewer's Responses to Questions 

Comments to the Author

1. Is the manuscript technically sound, and do the data support the conclusions? The manuscript must describe a technically sound piece of scientific research with data that supports the conclusions. Experiments must have been conducted rigorously, with appropriate controls, replication, and sample sizes. The conclusions must be drawn appropriately based on the data presented. 

Reviewer #1: Partly 

Reviewer #2: Yes 

2. Has the statistical analysis been performed appropriately and rigorously?

Reviewer #1: Yes 

Reviewer #2: Yes

3. Have the authors made all data underlying the findings in their manuscript fully available? The PLOS Data policy requires authors to make all data underlying the findings described in their manuscript fully available without restriction, with rare exception (please refer to the Data Availability Statement in the manuscript PDF file). The data should be provided as part of the manuscript or its supporting information, or deposited to a public repository. For example, in addition to summary statistics, the data points behind means, medians and variance measures should be available. If there are restrictions on publicly sharing data—e.g. participant privacy or use of data from a third party—those must be specified.

Reviewer #1: Yes 

Reviewer #2: Yes

4. Is the manuscript presented in an intelligible fashion and written in standard English? PLOS ONE does not copyedit accepted manuscripts, so the language in submitted articles must be clear, correct, and unambiguous. Any typographical or grammatical errors should be corrected at revision, so please note any specific errors here.

Reviewer #1: No

Reviewer #2: No

5. Review Comments to the Author

Reviewer 1:

In first place, I would like to congratulate the authors for conducting a study evaluating the relationship between adverse maternal outcomes and adverse neonatal outcomes in operative vaginal deliveries, as it is a clinically relevant topic that requires research in order to improve health assistance.

The title ‘Unfavorable Feto-Maternal Outcomes and Associated Factors of Operative Vaginal Delivery among Women Delivered in East Gojjam Zone Hospitals, North West Ethiopia’ describes correctly the purpose of the study.

I believe the study has several highlights:

- It is a multicenter study carried out in nine different hospitals, reaching a consecutive sample of pregnant women who delivered by either forceps or vacuum in a short perio of time (March 1 to April 30, 2019).

- Statistical design including sample size calculation and sampling procedure is well described in the ‘Methods’.

- Analysis of possible risk factors associated with adverse maternal and neonatal outcomes is thoroughly described.

- Tables 3 and 4 are nicely presented and easy to understand.

However there are some issues I believe the authors must address.

MAJOR ISSUES:

- the English language used is confusing and makes the manuscript difficult to follow and understand. I suggest a thorough linguistic revision by native English speaker so that the paper is more approachable and other aspects of the study can be addressed.

- The title, although describes the main idea of the study is long and confusing. I recommend reviewing.

-The abstract does not include the main objective of the study in a separate subsection. Within the ‘Background’, the authors state: ‘Furthermore, no study reported both maternal and neonatal outcomes at a time.’ Are the authors certain of this statement?

- Introduction:

Authors state (lines 56-58): ‘Neonatal trauma is associated with initial unsuccessful attempts at operative vaginal delivery by inexperienced operators.’ Are the authors certain of this statement? Could they provide reference?

- Objective of the study is not clearly described.

- Methods:

This section is divided into multiple subsections (Study area, design and period; Study design; Source population; Inclusion and exclusion criteria; Sample size and sampling procedure; Sampling procedure; Study variables; Operational definitions; Data quality-control; Daa processing and Analysis and Ethical clearance). These sections make the ‘Methods’ of the study unclear and confusing. Furthermore, similar information is repetead in the different subsections, for example ‘Source population’ and ‘Study population’

Sampling procedure: Authors state that mothers and neonates were followed for treatment and obstetrical care outcomes until discharged (six hours) from each selected hospital. I believe six hours is a very short period of time to report accurately the rate of adverse maternal and neonatal outcomes.

Are subsequent Emergency visits to the hospital after discharge evaluated in these patients?

- Results:

I suggest changing ‘Result’ for ‘Results’.

In this particular section, English is really confusing, making it really difficult to follow the results the authors obtain (example: Line 215 ‘Greater than three fourth (77%) of the respondents were…’).

Terms such as ‘asphyxia’ should be well defined before being included in the results section.

Some phrases are repeated practically identically in different paragraphs (example: Line 242: Finally, educational status, residence, sex of the neonate …..’. Line 246 Educational level […], resident […]’)

- Discussion:

Again, English used throughout the section is confusing, being difficult to understand what the authors are trying to discuss.

References are missing in order to provide evidence for certain statements such as (Line 289): The reason might be male fetus has the probability of being post-term pregnancy and as a result weight might be increased than female fetus which increase the bad maternal outcome of operative vaginal delivery.’

- Limitation:

Only one limitation is stated in this section. I believe the study has several more limitations that the authors should state in this section.

- Conclusion:

The first sentence of the ‘Conclusion’ is confusing: This study revealed that the overall unfavorable maternal and neonatal outcomes of operative delivery are found to be higher in East Gojjam Zone hospitals.’ Higher than what?

- References do not always respect Vancouver style.

- Tables:

Table 2 is too long and difficult to follow.

Reviewer 2:

Overall, while it appears that the paper is written in logical progression, with a background that explains its importance fairly well, it is inescapable that the scientific writing is not quite at the level necessary for publication. I suggest using the free online service “Grammarly” to run your paper through which will adjust for minor grammatical errors that are interfering with the flow and delivery of your message. I believe that PlosOne benefits greatly from including research from an international audience for a variety of reasons, but this manuscript as it stands is not quite ready for publication. I fully encourage your group to continue this line of inquiry and make the minor adjustments needed in your manuscript for PlosOne to reconsider it.

Abstract:

Line 10: Instead of proved, consider a word like demonstrated as that is merely the interpretation of a statistic, not inherently proof.

Line 14: define operative vaginal delivery briefly at this point

Line 21: Multivariable what? Define type of statistic further (or merely mention you are referring to statistics)

Results: typically an AOR needs to be done against specific variables that are controlled against. It appears that is approached, but not fully explained. If done so later, it is fine.

Conclusion: Conclusion needs to be more specific than “were high”. What was the true interpretation of the adjusted statistics?

Introduction:

Line 39: Obstetricians

Line 40: Difficulty deciding whether or not to intervene in the

Line 43: Why is the rate in Canada relevant for you? If you keep this fact, make it clear this is merely an example.

Line 47: instead of / write “and”

Line 56: Transition needed

Methods:

Note that populations statistics are rough estimates.

Great job noting hospital density

Who collected data and how were patients selected? – points to other potential confounding factors

Study Variables: perhaps better demonstrated in a table with further information and statistics?

Results and discussion are quite good. Minor grammatical changes but science and structure is well thought out.

I look forward to reading this manuscript again soon.

6. PLOS authors have the option to publish the peer review history of their article (what does this mean?). If published, this will include your full peer review and any attached files.

Do you want your identity to be public for this peer review? For information about this choice, including consent withdrawal, please see our Privacy Policy. 

Reviewer #1: No

Reviewer #2: No 

---

## [Author Response · Author response to Decision Letter 0]

29 Jan 2022

Thank you for your constructive comments. Here below are some responses for your questions 

1. ‘Furthermore, no study reported both maternal and neonatal outcomes at a time.’ Are the authors certain of this statement?

Yes we have searched a lot but we could not find similar study in the study area.

2. Methods

We prepared the manuscript based on the journal guideline. 

source and study population were not similar(study population were only population during the study period)

3. Are subsequent Emergency visits to the hospital after discharge evaluated in these patients?

We didn’t visit after discharge because our objective was to assess immediate maternal and neonatal outcome.

4. Terms such as ‘asphyxia’ should be well defined before being included in the results section

We operationalized terms that had different meaning in the study. Asphyxia is scientifically known term had no different meaning in the study.

5. Only one limitation is stated in this section. I believe the study has several more limitations that the authors should state in this section.

Based on our view we could not get any limitation other than the limitation we wrote.

---

## [Editor Report · Decision Letter 1]

9 May 2022

Immediate Unfavorable Birth outcomes and determinants of Operative Vaginal Delivery among Mothers Delivered in East Gojjam Zone Public Hospitals, North West Ethiopia: a cross-sectional study

PONE-D-21-32649R1

Dear Dr. Aynalem,

We’re pleased to inform you that your manuscript has been judged scientifically suitable for publication and will be formally accepted for publication once it meets all outstanding technical requirements.

Kind regards,

Sherif A. Shazly, M.B.B.Ch

Academic Editor

PLOS ONE

---

## [Editor Report · Acceptance letter]

23 May 2022

PONE-D-21-32649R1 

Immediate Unfavorable Birth outcomes and determinants of Operative Vaginal Delivery among Mothers Delivered in East Gojjam Zone Public Hospitals, North West Ethiopia: a cross-sectional study 

Dear Dr. Aynalem:

I'm pleased to inform you that your manuscript has been deemed suitable for publication in PLOS ONE. Congratulations! Your manuscript is now with our production department. 

Kind regards, 

on behalf of

Dr. Sherif A. Shazly 

Academic Editor

PLOS ONE